# Adherence Kinetics of a PDMS Gripper with Inherent Surface Tackiness

**DOI:** 10.3390/polym12112440

**Published:** 2020-10-22

**Authors:** Umut D. Çakmak, Michael Fischlschweiger, Ingrid Graz, Zoltán Major

**Affiliations:** 1Institute of Polymer Product Engineering, Johannes Kepler University Linz, Altenbergerstrasse 69, 4040 Linz, Austria; zoltan.major@jku.at; 2Chair of Technical Thermodynamics and Energy Efficient Material Treatment, Clausthal University of Technology, Agricolastrasse 4, 38678 Clausthal-Zellerfeld, Germany; michael.fischlschweiger@tu-clausthal.de; 3School of Education, Johannes Kepler University Linz, Altenbergerstrasse 69, 4040 Linz, Austria; ingrid.graz@jku.at

**Keywords:** silicon rubber (PDMS), dynamic thermomechanical analyses, storage modulus, mechanical loss factor, viscoelastic gripper

## Abstract

Damage and fiber misalignment of woven fabrics during discontinuous polymer processing remain challenging. To overcome these obstacles, a promising switchable elastomeric adherence gripper is introduced here. The inherent surface tackiness is utilized for picking and placing large sheets. Due to the elastomer’s viscoelastic material behavior, the surface properties depend on loading speed and temperature. Different peeling speeds result in different adherence strength of an interface between the gripper and the substrate. This feature was studied in a carefully designed experimental test set-up including dynamic thermomechanical, as well as dynamic mechanical compression analyses, and adherence tests. Special emphases were given to the analyses of the applicability as well as the limitation of the viscoelastic gripper and the empirically modeling of the gripper’s pulling speed-dependent adherence characteristic. Two formulations of poly(dimethylsiloxane) (PDMS) with different hardnesses were prepared and analyzed in terms of their applicability as gripper. The main insights of the analyses are that the frequency dependency of the loss factor tanδ is of particular importance for the application along with the inherent surface tackiness and the low sensitivity of the storage modulus to pulling speed variations. The PDMS-soft material formulation exhibits the ideal material behavior for an adhesive gripper. Its tanδ varies within the application relevant loading speeds between 0.1 and 0.55; while the PDMS-hard formulation reveals a narrower tanδ range between 0.09 and 0.19. Furthermore, an empirical model of the pulling speed-dependent strain energy release rate G(v) was derived based on the experimental data of the viscoelastic characterizations and the probe tack tests. The proposed model can be utilized to predict the maximum mass (weight-force) of an object that can be lifted by the gripper

## 1. Introduction

The picking and placing of limp solids (textile, woven fabrics, soft, and flexible electronics) during assembling in manufacturing processes remain challenging. The transfer to the desired position has to be achieved without damaging or misaligning the limp solid (i.e., fiber unravel, pull out, contamination by the gripper, etc.). A variety of technologies are available to overcome this challenge including vacuum suction, ingressive and adhesive grippers. The first two methods directly interact with the substrate’s surface to be transferred and deform it to some extent: vacuum suction results in a local lift up of the limp substrate and ingressive picking in penetrating as well as interlocking with fibers. An adhesive gripper, on the other hand, adheres on the surface, and the substrate itself is neither deformed nor misaligned. The adhesive part is usually an elastomer (e.g., copolymers and plasticized elastomers) with permanently tacky surface [1,2]. A major drawback is that the adherence cannot be switched off and so the release of a substrate is critical for a textile or soft electronic part. The modification of surface properties and specifically the enhancement of the adsorption abilities (e.g., for catalysts) [3,4,5,6,7] and can help to achieve switchable grippers. Alternatively, transfer printing is utilized in the assembly of micro/nanofabrication, where a silicon elastomer stamp with a viscoelastic surface behavior is used to deliver a part from a donor to a receiver substrate (see, e.g., Cheng et al. (2012) [8]). At the interface of two elastic materials the adhesion force is rather a constant than tunable [9]. However, the adhesion can depend on the peeling speed (high → lifting and low → release), the mechanical loading protocol (directional shearing induced delamination), temperature (laser pulse) or could be controlled by the surface relief structure as it was pointed out by Cheng et al. (2012) [8], Li et al. (2012) [10], and Chen et al. (2013) [9]. Extensive research efforts were presented to exploit the switchable adhesion in kinetically controlled [9,11,12], shear-assisted [8,13], direction-controlled [14], laser-driven non-contact [10,15], and microstructure enabled transfer printing [16,17,18,19]. Furthermore, in robotics various designs for gripper are presented in order to pick up and handle arbitrary objects (see, e.g., Brown et al. (2010) [20]). This influence of external physical loadings on the elastomeric gripper is exploited in order to achieve a controlled picking and placing of a substrate. The pronounced viscoelastic behavior of the gripper with a high loading rate sensitivity of the mechanical properties is therefore important. 

However, up to now no detailed study on the viscoelastic characteristics of an elastomeric adhesive gripper is presented. A throughout understanding of the interplay between loading rate sensitivity of the gripper’s bulk property and the inherent surface tackiness is of particular necessity to optimize the performance of an adhesive gripper. This paper presents an approach to gain insights to the temperature and loading rate dependent storage and dissipative (loss) properties by dynamic thermomechanical analyses of adhesive grippers as well as probe tack tests. The grippers under investigation are made of two different formulations of poly(dimethylsiloxane). Two woven fabrics (limp substrate) and two woven reinforced thermoplastic matrix composites (stiff substrate) are used to examine the gripper capabilities by the probe tack test. In “Material and Specimens”, the material selection is briefly discussed and the specimen geometries for the mechanical characterization (“Experimental”) presented. The loading rate dependent bulk and interface properties are modeled by simple linear viscoelastic theory assumptions and linear elastic fracture mechanics approach (“Experimental”). Based on the comprehensive experimental characterization the main mechanical features required for a viscoelastic adhesive gripper is summarized and presented in “Results and Discussion”.

## 2. Materials and Specimens

The silicone rubbers (poly(dimethylsiloxane); PDMS) under investigation were mixed and cured with two different base polymer:catalyst ratios (PDMS-soft → 20:1 and PDMS-hard → 10:1). The volume fraction of catalyst, curing temperature, and time to achieve a solid material are thereby critical, whose surface exhibits tackiness [21] high enough to pick up fabrics from a stack; the higher the tackiness the higher is the risk of contamination in a repetitive assembly process. Details about the fabrication of the material can be found in Cakmak et al. (2014a) [22].

Two categories of specimens were prepared. One category was dedicated for the general mechanical characterization of PDMS by dynamic thermomechanical analysis (DTMA); the other was for component tests under an application-related measurement protocol as well as dynamic mechanical analysis (DMA). DTMA was performed with the “barbell” specimen and the component tests were investigated with a cylindrical stamp. Figure 1 shows the specimen geometries and the respective dimensions.

In the following section, the experimental procedure (“Experimental”) for the characterization of the employed PDMS is presented. Furthermore, the design and the application-related experiments of the gripper stamp are reported there.

## 3. Experimental

After the initiation of the idea to adopt the rate and delamination loading-dependent (switchable) adherence from transfer printing to a large-scale application of limp solids (fabrics), some promising preliminary experiments were performed. Motivated by these preliminary results, an experimental scheme limited to mechanical characterizations only was defined. All experiments were performed with an electro-mechanic/dynamic actuator (TestBench, Bose Corp., ElectroForce Systems Group, MN, USA).

### 3.1. Dynamic (Thermo-) Mechanical Analysis (D(T)MA)

In order to gain more information concerning the viscoelastic nature of the PDMS, D(T)MA experiments were performed with the barbell specimen as well as the gripper stamp. A classical D(T)MA test procedure was defined including frequency dependent experiments from 1 Hz to 16 Hz under isothermal condition at seven different temperatures (−30 to +30 °C). The loading mode was uniaxial tensile with a sinusoidal excitation of a mean strain level of 1% and a dynamic amplitude of 0.5%. A similar test procedure was performed with the gripper stamp in order to determine the viscoelastic behavior of the PDMS-soft material under component relevant condition. Component relevant condition means, thereby, a uniaxial compression dynamic mechanical analysis (DMA) just before delamination from the substrate. Compression DMA was examined at room temperature (22 °C) with a mean level of −0.5 mm and amplitude of 0.5 mm excitation from 0.1 Hz to 47 Hz. The data of the D(T)MA experiments were analyzed in WinTest software (Bose Corp., ElectroForce Systems Group, MN, USA) and the storage (*E’*) as well as the transient (*E’’* and tanδ = *E’’*/*E’*) mechanical material properties were exported for further analyses. The complex modulus E* is given by Equation (1) and can be modeled by the well-known Prony series in the frequency (ω = 2πf) domain [23]. In the series *E*_0_ refer to the instantaneous modulus, *τ_i_* is the relaxation time and *g_i_* = (*E*_*i*−1_−*E_i_*)/*E*_0_ the Prony series coefficient corresponding to the relaxation time.
(1)$$E^{*}\left(\omega \right)=E^{\prime }\left(\omega \right)+j\cdot E^{\prime \prime }\left(\omega \right)=E_{0}\cdot \left(1-\sum_{i}\frac{g_{i}}{1+{\left(\tau_{i}\cdot a_{T}\omega \right)}^{2}}+j\cdot \underset{i}{\sum }\frac{g_{i}\tau_{i}a_{T}\omega }{1+{\left(\tau_{i}\cdot a_{T}\omega \right)}^{2}}\right)$$

If the material’s thermorheological behavior is simple, then the time–temperature superposition can be applied and resulting in master curve at a reference temperature [22,24]. The temperature dependent shift factor a_T_ can be given by the well-known WLF function [25] at a certain temperature *T* where *T*g is the glass transition temperature:(2)$$\left|log_{10}\left(a_{T}\right)\right|=17.44\cdot \frac{T-T_{g}}{51.6+T-T_{g}}$$

By inverse Laplace transformation of *E** into the time-domain, the relaxation modulus in time is obtained:(3)$$E\left(t\right)=E_{0}\cdot \left(1-\sum_{i}g_{i}\cdot \left(1-e^{-t/\tau_{i}}\right)\right)=E_{\infty }\cdot \frac{1-\sum_{i}g_{i}\cdot \left(1-e^{-t/\tau_{i}}\right)}{1-\sum_{i}g_{i}}$$
where $E_{\infty }$ = *E*_0_ ($1-\underset{i}{\sum }g_{i})\;$ is the Young’s modulus after infinite long time.

For further analyses of the rate-dependent adhesion, Equation (2) will be more convenient. From an experimental point of view, the DMA tests are preferable, as the storage and the transient viscoelastic behaviors are determined at once.

### 3.2. Probe Tack Test

The gripper stamp made of PDMS-soft was investigated in order to determine the rate dependent adherence properties for different substrate surfaces. Basically, the adherence test was a tensile delamination test (see Figure 2) at various pulling speeds (0.1 mm/s up to 100 mm/s) in accordance to the surface tack measurement set-up of Çakmak et al. (2011) [21]. The stamp was pressed onto the substrate, held for 10 s at −10 N, followed by the pulling and delamination from the substrate surface, while the force F was measured. In a real application, the stamp will lift up the substrate, but here the adherence was of particular interest and therefore the substrate was fixed. The substrates (100 × 100) mm^2^ were attached to the support by means of double-side adhesive tape. To avoid inertia effects at high pulling speeds, the force was measured at the fixed side of the test setup. The maximum force corresponds to the fully delaminated stamp and is the limiting factor in-service for the considered pulling speed. If components with higher weight forces are applied, the stamp is not capable to pick them up. Application-relevant substrates were investigated and comprised dry carbon and glass fiber woven fabrics as well as poly(amide) matrix organo sheets reinforced with carbon and glass fiber weave.

The delamination process is assumed to be driven mainly by the crack propagation from the edge of the stamp when a tractive force *F* was applied. For the sake of completeness, it should be mentioned that the other mechanisms would be the crack propagation starting from inside at some interfacial defects and decohesion at theoretical contact strength [26]. The real delamination process will be rather a combination of the aforementioned mechanisms. However, visual observations of the experiments revealed that the crack initiation and propagation at the edge was more dominant. From fracture mechanical point of view, the stress singularity at the crack tip can be sufficiently described by the stress intensity factor *K*_I_. When the stamp is fully propagated along the interface, the crack length is equal to the radius *R* of the stamp and *K*_I_ is given by
(4)$$K_{I}=\frac{F}{\pi \cdot R^{2}}\cdot \sqrt{\pi \cdot R}\cdot f$$
where *F* is the tractive force (*F*/π*R*^2^ is the tractive stress σ) and *f* is the geometry coefficient and for a pillar shaped geometry *f* = ½. The energy needed to create an area surface during a plain strain loading is connected to the stress intensity factor K_I_ by
(5)$$G=\frac{K_{I}^{2}\cdot \left(1-\nu^{2}\right)}{2\cdot E}$$
where *ν* is the Poisson’s ratio and *E* is the Young’s modulus. Since the material under investigation can be considered as incompressible (*ν* = 0.5) and by combining Equation (4) with (5), the following relation can be found.
(6)$$G_{c}\left(v,t_{s}\left(v\right)\right)=\frac{3\cdot \sigma {\left(v\right)}^{2}\cdot \pi \cdot R}{32\cdot E\left(t_{s}\right)}$$

Equation (6) shows the critical energy release rate per area depending on the pulling speed v and the separation time *t*_s_. The separation time is in turn a function of the pulling speed, meaning that the higher v, the lower *t*_s_. This dependency is mainly driven by the viscoelastic nature of the stamp material and is also examined experimentally. To determine the actual value of the Young’s Modulus *E*(*t*_s_), knowledge of the separation time is crucial. The pulling velocity-dependent tractive stress σ(v) has to be obtained by the component tests and is mainly determined by the surface energies of each solid in contact and the surface interaction energy of them in direct contact (cf. Duprè energy of adhesion). It is expected that the investigated substrates have a different trend in terms of measured tractive stresses. However, the velocity dependency of σ(v) is a respond given by the stamp’s viscoelastic nature and can be described by the following square root function sufficiently enough.
(7)$$\sigma \left(v\right)=\sigma_{0}\cdot \sqrt{1+{\left(\frac{v}{v_{0}}\right)}^{n}}$$

When the square root function is determined by simply fitting the measured data and combining Equations (3) and (7) with (6) lead to the pulling speed-dependent critical energy release rate:(8)$$G_{c}\left(v,t_{s}\left(v\right)\right)=G_{0}\cdot \frac{1+{\left(\frac{v}{v_{0}}\right)}^{n}\cdot \left(1-\sum_{i}^{3}g_{i}\right)}{1-\sum_{i}^{3}g_{i}\cdot \left(1-e^{-t_{s}/\tau_{r}}\right)}$$
where *G*_0_ is the critical energy release rate near zero pulling speed v and infinite separation time *t_s_*, *v*_0_ is the reference pulling speed and n is the scaling exponent. Equation (8) is in accordance with the empirical form shown and utilized by Shull (2002) [27] and Feng et al. (2007) [11]. The modification is related to the time dependent Young’s modulus and so the Prony series is incorporated. Parameter identification of the function in Equation (8) is carried out based on the experimental data of the component tests as well as DMA. The model will be used to predict the critical energy release rate for the desired in-service pulling speed.

## 4. Results and Discussion

As was mentioned in the “Materials and Specimens” section, two different formulations were considered for the application of the gripper stamp. These two materials will be herein after referred to as PDMS-soft and PDMS-hard. Basically, only the PDMS-soft material was suitable for the use as gripper; however, basic viscoelastic characterization (D(T)MA) was performed for both of the materials in order to show the range of possible mechanical behavior. Only PDMS-soft was investigated with the component testing procedure.

### 4.1. Dynamic (Thermo-) Mechanical Analysis (D(T)MA)

Figure 3 shows the DTMA results of the investigated materials. Each color corresponds to the results of isothermal testing at various frequencies, which are indicated as single points. 

This diagram is very convenient for analyzing the trend of the inherent respond time’s temperature dependency as a result of the external mechanical loading. If all examined data reveal that tanδ is a unique function of *E’*, then the respond times (here the relaxation times) are equally temperature-dependent and time–temperature superposition (TTSP) is applicable (cf. Çakmak and Major (2014b) [28] and Tschoegl et al. (2002) [24]). Besides that, PDMS-soft formulation’s characteristic is located within the diagram rather at the top right than the trend of PDMS-hard as expected. Important is that for both cases the TTSP can be applied, shift factors determined to construct the master curves at reference temperatures and these factors could be modeled with WLF function, among others.This is illustrated in Figure 4 for PDMS-soft (the candidate material for the application).

If the shift factor function (WLF) is applicable, then the temperature influence on the contact mechanical behavior can be easily predicted (e.g., Feng et al. (2007) [11]). This is of particular interest when the environmental conditions in-service of the gripper (i.e., during manufacturing of products) are fluctuating.

Figure 5 demonstrates the compression DMA results as before illustrated for the DTMA results. Now the gripper stamps made of the PDMS-formulations are investigated under contact and mechanically excited before delamination occurs. As mentioned earlier, the surface tackiness is an important requirement for the gripper stamp. However, the analyses of the viscoelastic behavior reveal that the loss factor characteristics are equally significant. It is believed that a high frequency (loading rate) dependency is essential to exploit the rate dependent adhesion. If the material has almost no rate-dependent tanδ, which is the case for PDMS-hard formulation, then the material will not be suitable as gripper with rate dependent adhesion. The storage modulus and its rate dependency are of minor importance. It should be low and remain low with increasing loading rate.

The storage modulus of the PDMS-soft material is modeled with the 3rd order Prony series (cf. Equation (1)) and the respective Prony parameters are listed in Table 1. Figure 6 shows the experimental data of the frequency dependent storage modulus and the mentioned fit curve. 

With the Prony series, the time dependent storage modulus needed for computing and modeling the critical energy release rate is obtained.

### 4.2. Probe Tack Test

From the tack tests the tractive stresses are evaluated, which are measured as the maximum stresses before delamination at various pulling speeds and are shown in Figure 7. The experimental data confirm the assumption of the square root relation between the stresses and the pulling speeds shown in Equation (7). This function’s fit parameters for the investigated substrates are given in Table 2.

With *E*_∞_ and σ_0_, the critical energy release rate G_0_ near zero pulling speed and infinite separation time can be calculated. Table 2 shows the computed values and it can be observed, that the fabrics have a similar G_0_ as well as the organo sheets. The surfaces of the fabrics are comprised by the fibers and the empty space in between and so the surface adhesion is determined by this meso-structure; on the other hand, the organo sheets’ surfaces are dominated by the thermoplastic PA6.

Moreover, the time to separate the gripper stamp from the fixed substrates is evaluated (see Figure 8) in order to determine the actual storage modulus according to Equation (3). Figure 8 reveals that the separation time is with a good approximation independent of the substrate in contact with the gripper stamp. The red line shows the exponential decay trend over the pulling speed.

From the data presented above, the critical energy release rates G are computed according to Equation (6) and are shown to be pulling speed-dependent in Figure 9. Also the modeled energy release rates are shown with good agreement with the experimental data. The model seems to be appropriate to model and predict the critical energy release rate for in-service use. From this, the pulling speed needed to lift up/release an object with a certain mass could be calculated as well.

## 5. Conclusions

The presented methodology to evaluate the viscoelastic as well as surface tack characteristics revealed that the gripper made of the soft PDMS formulation is capable for grasping, holding, transporting, and finally assembling (draping) of fabrics as well as semi-finished thermoplastic parts (organo sheets with PA6 matrix). Based on the observations and the experimental data, the essential viscoelastic as well as adherence kinetic behaviors are determined. A high loading speed dependency of the tanδ characteristic is needed (i.e., with higher loading speed, higher tanδ) to exploit the rate-dependent adherence. The tanδ of PDMS-soft varies between 0.1 (@0.1 Hz) and 0.55 (@46.5 Hz), while the PDMS-hard formulation reveals a narrower tanδ range between 0.09 and 0.19. If the material has almost no rate-dependent tanδ, which is the case for PDMS-hard formulation (see Figure 5), then the material is inapplicable as a gripper. The storage modulus and its loading speed dependency are of minor importance (PDMS-soft: 0.25 MPa (@0.1 Hz) to 0.5MPa (@46.5 Hz); PDMS-hard: 2.4 MPa (@0.1 Hz) to 4.5 MPa (@46.5 Hz)). It should be low and remain low with increasing loading speed.

Grasping and assembling are achieved by the utilization of the rate-dependent adhesion. For the application, the knowledge of the gripper’s pulling speed dependent adherence characteristic is essential. Here, a model (see Equation (8)) is derived based on the experimental results and describes sufficiently the rate dependency of the gripper. It can be applied for the prediction of the critical energy release rate and, more importantly, the maximum mass (weight force) to lift and hold. We believe that the proposed model is of particular interest for the application and operation of a robot handling system with adhesive grippers. In the presented work, the aging, and therefore the altering of the gripper material’s mechanical inherent viscoelastic properties, are omitted. For the prediction of the service (replacement) interval times, the aging behavior has to be known and has to be verified in future.

## Figures and Tables

**Figure 1 polymers-12-02440-f001:**
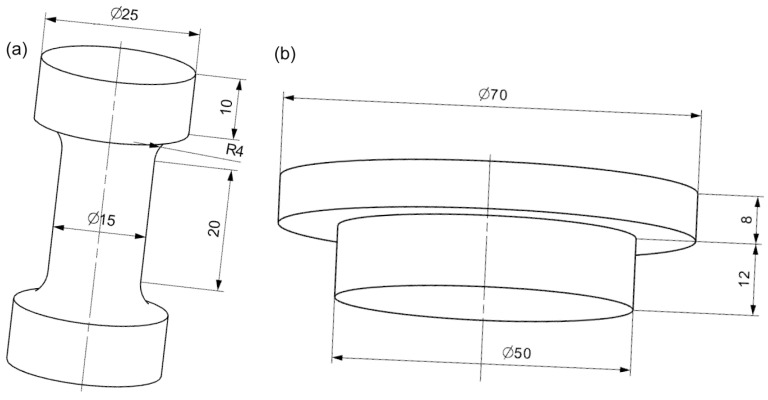
Specimen geometries: (**a**) barbell specimen and (**b**) gripper stamp specimen. Dimensions are in mm; R: radius.

**Figure 2 polymers-12-02440-f002:**
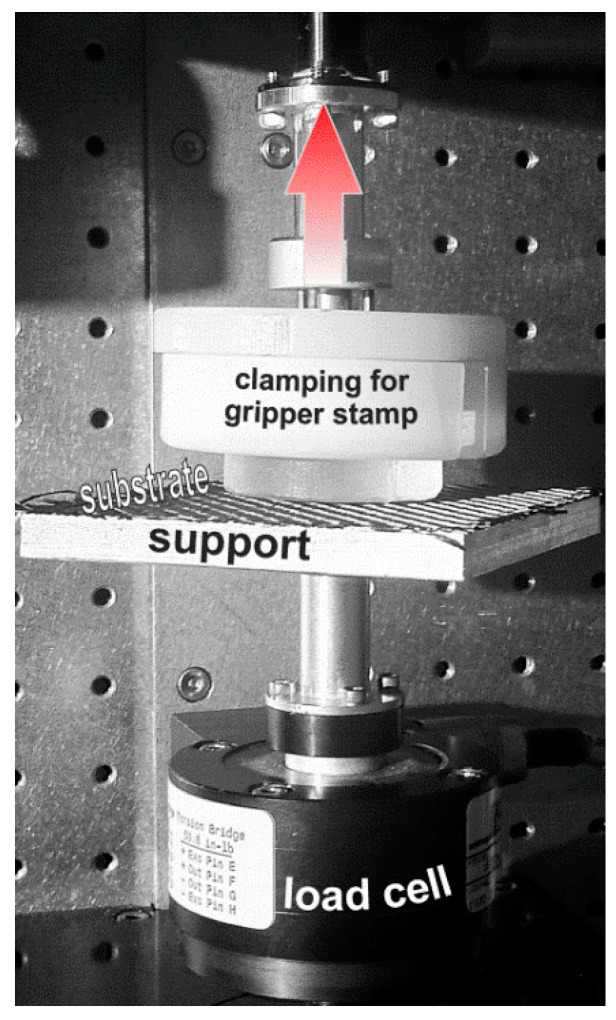
Component test set-up.

**Figure 3 polymers-12-02440-f003:**
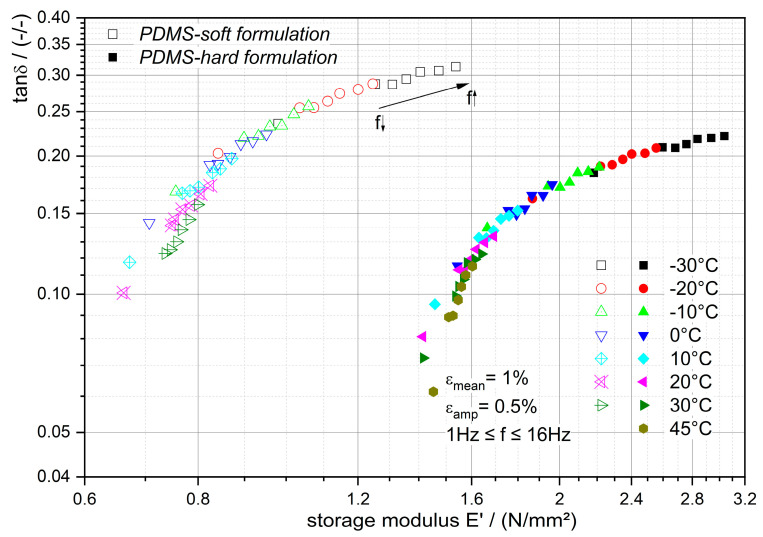
Loss factor tanδ over storage modulus *E’* characteristic of the investigated materials at various temperatures.

**Figure 4 polymers-12-02440-f004:**
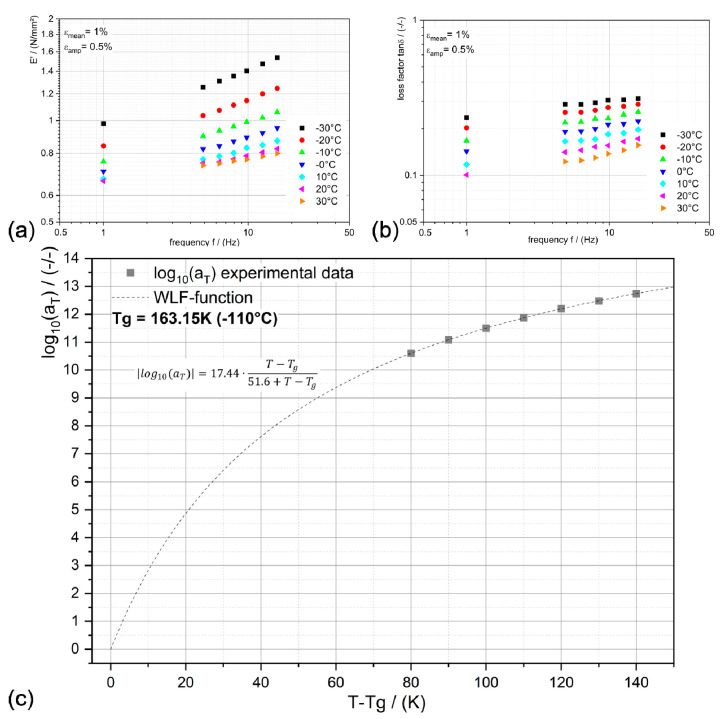
Panels (**a**,**b**) show the temperature and frequency dependent storage modulus and loss factor, respectively. In panel (**c**), the obtained time-temperature shift factors a_T_ are presented as points and the WLF function as dashed line.

**Figure 5 polymers-12-02440-f005:**
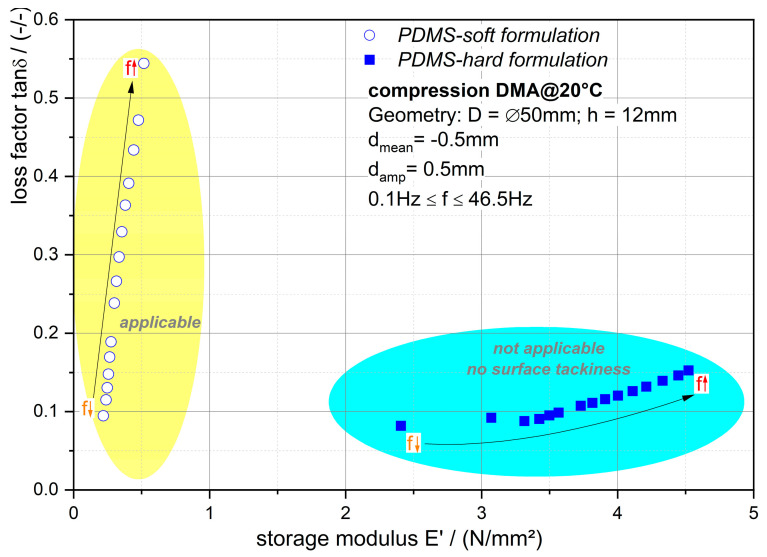
Loss factor tanδ over storage modulus *E’* characteristic of the investigated formulation at 20 °C under compression DMA loading procedure.

**Figure 6 polymers-12-02440-f006:**
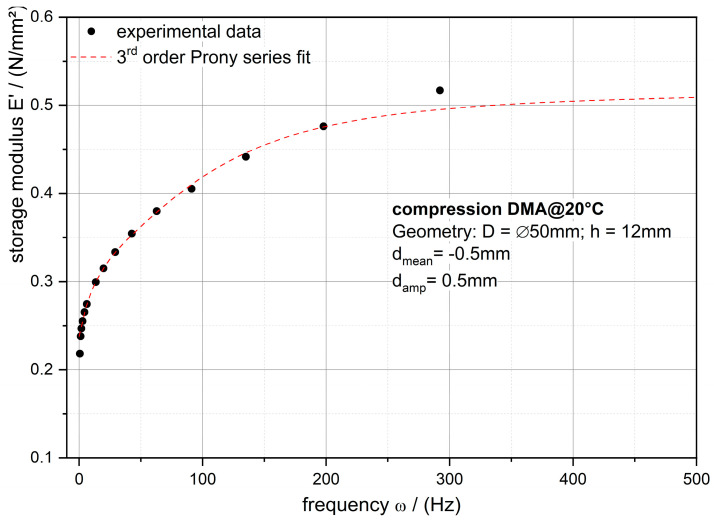
Linear plot of the frequency (ω = 2π)-dependent storage modulus *E’* of PDMS-soft formulation (black points) with the 3rd order Prony series fit of the data (cf. Table 1)**.**

**Figure 7 polymers-12-02440-f007:**
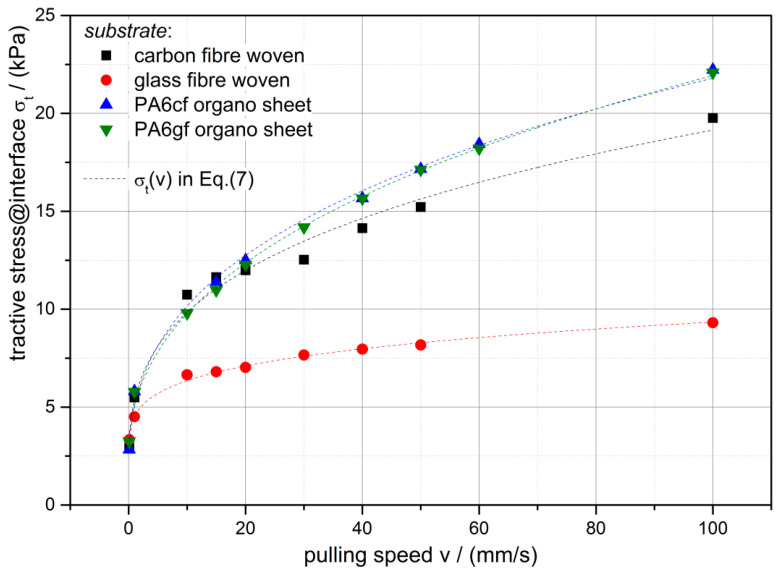
Measured tractive stresses σ_t_ between the PDMS-soft gripper stamp and the investigated substrates at various pulling speeds (points) with the fit function (Equation (7)) as dashed lines.

**Figure 8 polymers-12-02440-f008:**
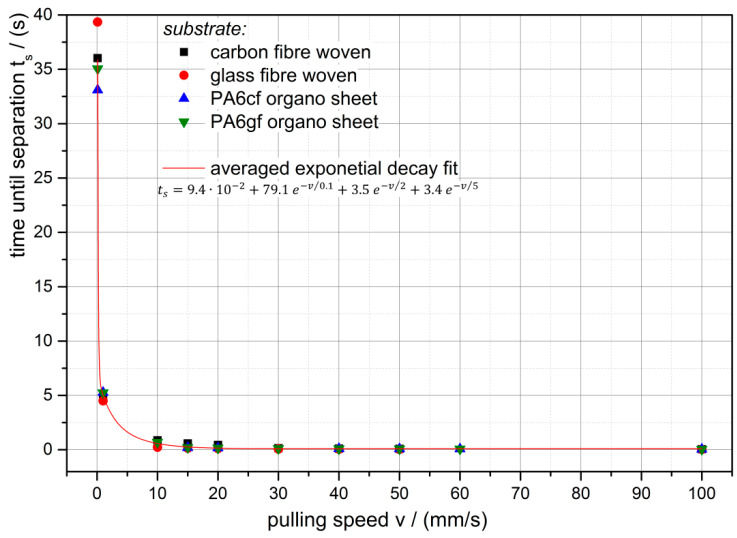
Pulling speed related time to separate the gripper stamp from the substrates; red line shows the trend of all measurement points.

**Figure 9 polymers-12-02440-f009:**
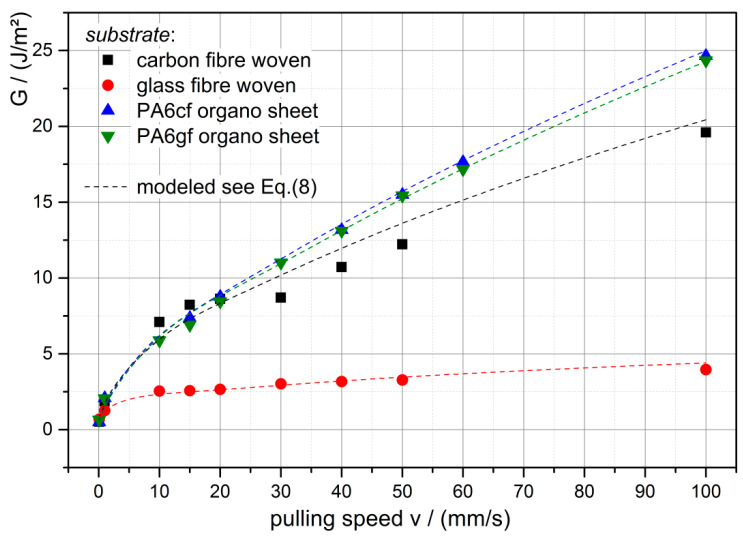
Critical energy release rate G versus the pulling speed v including the model of Equation (8) for each substrate.

**Table 1 polymers-12-02440-t001:** Prony parameters of PDMS-soft formulation.

Gripper Stamp	*E* _0_	*E* _∞_	*g* _i_	τ_i_
	/(MPa)	/(MPa)	-	/(s)
PDMS-soft	0.517	0.238	8.35 × 10^−2^	2.50 × 10^−1^
			1.12 × 10^−1^	5.23 × 10^−2^
			3.44 × 10^−1^	9.22 × 10^−3^

**Table 2 polymers-12-02440-t002:** Fit parameter of the tractive stress and the corresponding calculated G_0_.

Substrate	σ_0_	n	G_0_
v_0_ = 0.1 mm/s	/(kPa)	-	/(J/m^2^)
carbon fiber woven	2.4	0.60	0.36
glass fiber woven	2.5	0.38	0.38
PA6 carbon fiber	2.1	0.68	0.27
PA6 glass fiber	2.0	0.68	0.26

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
