# Peer review of "Adherence Kinetics of a PDMS Gripper with Inherent Surface Tackiness"

_polymers, 2020, doi:10.3390/polym12112440_

Round 1

Reviewer 1 Report

The authors have presented a detailed study on the viscoelastic characteristics of an elastomeric adhesive gripper. Based on dynamic mechanical analysis (DMA) and an empirical model has been proposed. The manuscript has high novelty and scientific quality. The subject of this paper should be of interest for the readers of the Journal, the manuscript is acceptable for publication after a minor revision outlined below.

Minor comments:

- Page 2 line 67. The definition of the acronym PDMS should be given.

- Page 3 Figure 1. What does R4 mean?

- Page 3 line 86-87. I can’t find the predicate in this sentence.

- Page 3 line 102. It would be better to use one kind of acronym for dynamic thermomechanical analysis, DTMA or D(T)MA.

- Page 4. To be complete, T and Tg should be defined in eq. 2.

- Page 7 Figure 4. For the sake of clarity, it should be mentioned that the figure is related to PDMS-soft.

Author Response

Thank you very much for your comments! Your comments are modified in the revision.  Also the writing was improved. We believe that the modified abstract and conclusion now give more information and improve the quality of the manuscript.

Response to Reviewer #1:

The authors have presented a detailed study on the viscoelastic characteristics of an elastomeric adhesive gripper. Based on dynamic mechanical analysis (DMA) and an empirical model has been proposed. The manuscript has high novelty and scientific quality. The subject of this paper should be of interest for the readers of the Journal, the manuscript is acceptable for publication after a minor revision outlined below.

Minor comments:

- Page 2 line 67. The definition of the acronym PDMS should be given.

Response: poly(dimethylsiloxane) is included

- Page 3 Figure 1. What does R4 mean?

Response: R…radius is included in the caption of the figure

- Page 3 line 86-87. I can’t find the predicate in this sentence.

Response: “After the initiation of the idea to adopt the rate and delamination loading dependent (switchable) adherence from transfer printing to a large-scale application of limp solids (fabrics), some promising preliminary experiments were performed.” Is corrected

- Page 3 line 102. It would be better to use one kind of acronym for dynamic thermomechanical analysis, DTMA or D(T)MA.

Response: we decided to use D(T)MA

- Page 4. To be complete, T and Tg should be defined in eq. 2.

Response: Is now included as “…at a certain temperature T where Tg is the glass transition temperature:”

- Page 7 Figure 4. For the sake of clarity, it should be mentioned that the figure is related to PDMS-soft.

Response: at line 211 it is now mentioned: “This is illustrated in Figure 4 for PDMS-soft (the candidate material for the application).”

Reviewer 2 Report

In this manuscript, a promising switchable elastomeric adherence gripper is introduced. Some interesting results are presented. However, major revision is needed to further improve the manuscript.

For instance,The abstract and conclusion should be improved, and some values should be added.

In the introduction part, the authors should summarize the related attempts on the recent development of composites with improved performance. It would be necessary to discuss by referring some articles, such as: J Manuf Process, 2019,45: 520-531.; Environ Sci Technol, 2019, 53: 6989-6996; Appl Clay Sci,2018,152:221-229.; ACS Catal,2020,10:7894-7906.; Bioresource Technol, 2018,268:726-732.

The scientific principles should be further concluded in this work, and some pictures should be clearer.

There are some English language errors, which should be improved carefully.

Author Response

Thank you for your comments! We considered all your comments carefully and did the following revisions:

-) abstract is now longer and includes results

-) conclusion provides more details and comprise the main insights of the study and highlights the limitations including an outlook for future analyses.

-) the introduction includes the suggested improvements of surface properties

-) the writing was improved

We believe that the manuscript is now in a better quality with the revisions done.